

# Metal artifact reduction combined with deep learning image reconstruction algorithm for CT image quality optimization: a phantom study

Huachun Zou[1,2], Zonghuo Wang[2], Mengya Guo[3], Kun Peng[2], Jian Zhou[2], Lili Zhou[1] and Bing Fan[2]

[1] School of Medical and Information Engineering, Gannan Medical University, Ganzhou, China
[2] Department of Radiology, Jiangxi Provincial People's Hospital, The First Affiliated Hospital of Nanchang Medical College, Nanchang, China
[3] CT Imaging Research Center, GE Healthcare China, Beijing, China

## ABSTRACT

**Background**. Aiming to evaluate the effects of the smart metal artifact reduction (MAR) algorithm and combinations of various scanning parameters, including radiation dose levels, tube voltage, and reconstruction algorithms, on metal artifact reduction and overall image quality, to identify the optimal protocol for clinical application.

**Methods**. A phantom with a pacemaker was examined using standard dose (effective dose (ED): 3 mSv) and low dose (ED: 0.5 mSv), with three scan voltages (70, 100, and 120 kVp) selected for each dose. Raw data were reconstructed using 50% adaptive statistical iterative reconstruction-V (ASIR-V), ASIR-V with MAR, high-strength deep learning image reconstruction (DLIR-H), and DLIR-H with MAR. Quantitative analyses (artifact index (AI), noise, signal-to-noise ratio (SNR) of artifact-impaired pulmonary nodules (PNs), and noise power spectrum (NPS) of artifact-free regions) and qualitative evaluation were performed.

**Results**. Quantitatively, the deep learning image recognition (DLIR) algorithm or high tube voltages exhibited lower noise compared to the ASIR-V or low tube voltages ($p < 0.001$). AI of images with MAR or high tube voltages was significantly lower than that of images without MAR or low tube voltages ($p < 0.001$). No significant difference was observed in AI between low-dose images with 120 kVp DLIR-H MAR and standard-dose images with 70 kVp ASIR-V MAR ($p = 0.143$). Only the 70 kVp 3 mSv protocol demonstrated statistically significant differences in SNR for artifact-impaired PNs ($p = 0.041$). The $f_{peak}$ and $f_{avg}$ values were similar across various scenarios, indicating that the MAR algorithm did not alter the image texture in artifact-free regions. The qualitative results of the extent of metal artifacts, the confidence in diagnosing artifact-impaired PNs, and the overall image quality were generally consistent with the quantitative results.

**Conclusion**. The MAR algorithm combined with DLIR-H can reduce metal artifacts and enhance the overall image quality, particularly at high kVp tube voltages.

Corresponding authors
Lili Zhou, lilizhou369@gmu.edu.cn
Bing Fan, 26171381@qq.com

## INTRODUCTION

With the accelerated global aging population, metal implants for fixation or prosthetic replacement—including dental prostheses (*Bayerl et al., 2023*), spinal screws (*Enache et al., 2025*), hip arthroplasty (*Zhao et al., 2023*), and cardiovascular implantable electronic devices (CIEDs) (*Wong & Devereaux, 2019*)—have been extensively utilized. These metallic devices induce substantial imaging artifacts through multiple physical mechanisms such as photon starvation phenomena, beam-hardening effects and scatter, which collectively degrade diagnostic image quality in CT. Specifically, metal artifacts generated by CIEDs during CT scanning significantly degrade visualization of adjacent anatomical structures, including mediastinal vasculature, lymph nodes, and parenchymal tissues (*Pennig et al., 2021*; *Zhao et al., 2023*). These artifacts critically compromise diagnostic accuracy in routine thoracic CT applications, particularly impacting lung cancer screening sensitivity, treatment planning, and therapeutic response assessment (*Kikuchi et al., 2020*).

In recent years, various technical strategies have been developed to mitigate metal artifacts in CT imaging, including optimization of acquisition parameters (increased tube voltage and tube current) (*Selles et al., 2024*), high-keV virtual monoenergetic imaging of spectral CT (*Laukamp et al., 2019*; *Bongers et al., 2015*; *Khodarahmi et al., 2018*; *Long et al., 2019*) and metal artifact reduction (MAR) algorithms (*e.g.*, projection completion MAR, iterative MAR) (*Lehti et al., 2020*; *Wichtmann et al., 2023*). Among these, MAR techniques have emerged as pivotal solutions due to their ability to correct abnormal X-ray attenuation profiles caused by metallic implants through either projection data compensation or image domain iterations (*Chae et al., 2020*; *Choo et al., 2021*; *Dunet et al., 2017*; *Kim et al., 2020*; *Kanani et al., 2022*; *Kovacs et al., 2018*). Specifically, projection-based MAR algorithm such as smart MAR (GE HealthCare, Chicago, IL, USA) synthesizes corrected projections using a combination of both the original and substitutive projection data, potentially inducing global alterations in the projection domain (*Fukugawa et al., 2022*). However, most existing research predominantly focuses on the artifact reduction in artifact-impeded areas, while less attention is paid to artifact-free regions, particularly in terms of image texture preservation.

Furthermore, in images with metal artifacts, it's important to consider not only the extent of artifacts but also the overall image quality, including image noise, contrast and textures preservation. These comprehensive image quality metrics, as well as artifact reduction, are influenced by a variety of parameters. For instance, tube voltage impacts both image contrast and artifacts degree (*Zhao et al., 2023*), while the radiation dose and reconstruction algorithm generally impact the image noise (*Szczykutowicz et al., 2021*) in general. However, in some literature, certain reconstruction algorithms have demonstrated significant potential to reduce beam hardening artifacts (*Fujita et al., 2023*; *Yasaka et al., 2017*). For instance, *Li et al. (2024)* demonstrated that the artificial intelligence iterative reconstruction (AIIR) algorithm can mitigate streak artifacts caused by irregular arm positioning, thus reducing the likelihood of misdiagnosis. With the advancement of artificial intelligence, deep learning image reconstruction (DLIR, TrueFidelity, GE Healthcare) algorithms have emerged. This is a vendor-specific, deep convolutional neural

network-based image reconstruction technique that is trained under supervision with millions of parameters simultaneously, in order to produce an output image similar to filtered back projection (FBP) (*Zhu et al., 2024*). Compared to adaptive statistical iterative reconstruction-V (ASIR-V), which applies advanced noise, physics, and object modeling, they can effectively balance noise, radiation dose, and image texture (*Yang et al., 2021*). Previous studies have demonstrated that DLIR has an excellent ability to improve image quality and reduce radiation doses in metal-free scenarios, including thoracic (*Jiang et al., 2022a*; *Zhao et al., 2022*; *Yao et al., 2022*), abdominal (*Jensen et al., 2022*; *Caruso et al., 2024*), and cerebral CT (*Jiang et al., 2022b*; *Jiang et al., 2024*). While in images with metal artifacts, as we know, only *Sun et al. (2024)* investigated the feasibility of metal artifact reduction in low-dose spinal CT for post-surgical children based on a combination of the MAR and DLIR algorithms.

This study therefore performed a pacemaker-embedded (a specific type of CIED) phantom experiment to systematically investigate the metal reduction and image quality improvement of the combination between MAR and DLIR algorithms under various scan conditions (different tube voltages and radiation doses). The dual objectives focus on (1) optimizing metal-implant CT protocols, while (2) pioneering the clinical implementation of DLIR in artifact management. The key innovations reside in the novel integration of DLIR with MAR across diverse radiation dose regimes, coupled with the first comprehensive assessment of image texture fidelity in artifact-free regions.

## METHODS AND MATERIALS

### Phantom

In this study, the Lungman chest phantom (Lungman ph-1, Kyoto Kagaku Inc., Japan) was utilized. The anatomical structures, including the trachea, pulmonary vessels, and mediastinum, were simulated using tissue substitutes. Thirteen spherical nodules (CT value = $-800$ HU, corresponding diameters = 12, 10, 8, and 5 mm; CT value = $-630$ HU, corresponding diameters = 12, 10, 8, 5, and 3 mm; CT value = 100 HU, corresponding diameters = 12, 10, 8, 5 mm) were randomly placed in the phantom using cotton. To investigate the impact of metal artifacts, a pacemaker was attached to the upper left of the chest phantom (Fig. 1A). Four non-solid pulmonary nodules (PNs) were obscured by streak artifacts (Fig. 1B).

### Image acquisition and reconstruction

All scans were conducted using a 256-row multidetector CT scanner (Revolution Apex CT, GE Healthcare). To investigate the impacts of various scanning conditions on metal artifact reduction, three tube voltages (70, 100, and 120 kVp), along with their corresponding tube currents, were selected to achieve standard (3 mSv) and low (0.5 mSv) effective doses. The remaining scanning parameters were fixed across all scanning scenarios, as follows: display field of view (DFOV) of 42 cm × 42 cm, a pitch of 0.992, a detector width of 80 mm, a rotation time of 0.8 s/r, and a slice thickness of 1.25 mm. Furthermore, all acquisitions were reconstructed using high strength DLIR (DLIR-H), DLIR-H with metal artifact reduction (DLIR-H MAR), 50% adaptive statistical iterative reconstruction-V (ASIR-V), and ASIR-V

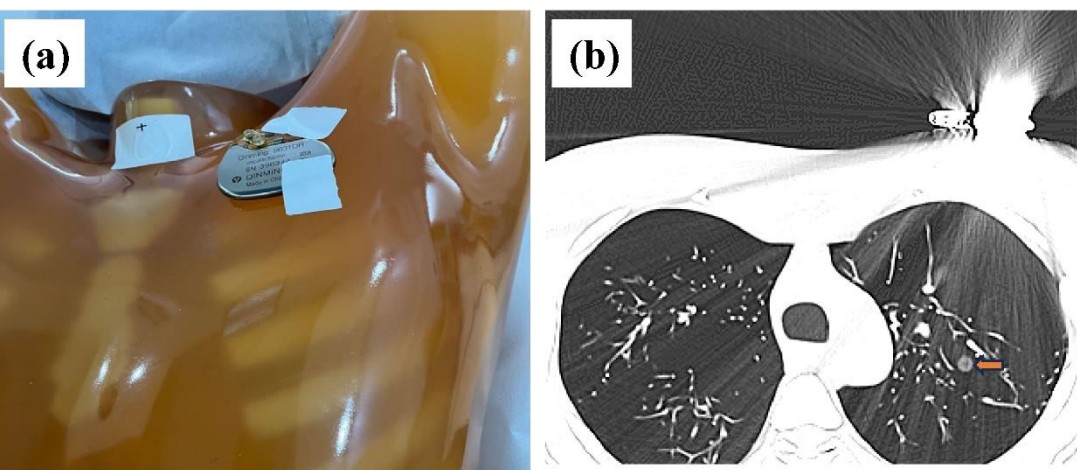

**Figure 1  Phantom configuration and corresponding CT images.** (A) The anthropomorphic thoracic phantom and implanted cardiac pacemaker. (B) Axial CT image with severe metal artifact impairment, where the artifact-impaired non-solid PNs are marked (yellow arrow).

with metal artifact reduction (ASIR-V MAR). CT scanning was repeated three times for each scenario.

## Objective image quality evaluation

For quantitative analysis, the artifact index (AI), background noise, signal-to-noise ratio (SNR) of artifact-impaired non-solid PNs, noise power spectrum (NPS) of artifact-free regions were calculated. All image sequences were loaded into MITK software (v2024.06, The German Cancer Research Center, Heidelberg, Germany), and region of interests (ROIs) were delineated in background air, artifact areas, and artifact-impaired PNs by a well-experienced radiologist (with nine years of chest radiology experience) based on 3 mSv 120 kVp DLIR-H MAR images. To ensure consistent quantitative analysis of the same ROIs in other images, these ROIs were saved and subsequently imported into other images for analysis. According to a previous study (*Chae et al., 2020*), the AI was quantified using the following formula: $AI = \sqrt{SD_{artifact}^2 - SD_{background}^2}$, where $SD_{artifact}$ and $SD_{background}$ represent the standard deviation (SD) of the streak artifact and background, respectively. To represent the extent of the artifacts as comprehensively as possible, ROIs ($50 \text{ mm}^2$) were placed in five consecutive artifact-pronounced slices. Background ROIs ($50 \text{ mm}^2$) were located in the air among five consecutive artifact-free slices, and the $SD_{background}$ in the AI formula was the average of these five background SD. To assess the influence of artifacts on PNs, the SNR values of artifact-impaired non-solid PNs (Fig. 1B) were calculated using the following formula: $SNR = \frac{|mean_{PN}|}{SD_{PN}}$, where $mean_{PN}$ and $SD_{PN}$ refer to the average and SD of the CT values of the four PNs at the maximum slice, respectively. To investigate the influence of DLIR and MAR algorithms on the image texture of artifact-free regions, NPS was evaluated in a homogeneous heart using imQuest software (Clinical Imaging Physics Group, Duke University, Durham, NC, USA) and the NPS area, average spatial frequency ($f_{avg}$) and peak spatial frequency ($f_{peak}$) were calculated.

### Subjective image quality evaluation

Two radiologists (5/9-year-experience in CT image diagnosis) independently performed subjective evaluations using a 5-point scale to assess the extent of metal artifacts, the confidence in diagnosing artifact-impaired PNs, and the overall image quality. The extent of metal artifacts was rated as follows: severe artifacts, unable to be diagnosed = 0, pronounced artifacts = 1, moderate artifacts = 2, mild artifacts = 3, and no artifacts = 4. The confidence in diagnosing artifact-impaired PNs was graded as follows: undetectable = 0, poorly detectable = 1, moderately detectable = 2, well detectable = 3, and manifestly detectable = 4. The overall image quality was rated as follows: very poor = 0, poor = 1, acceptable = 2, good = 3, and excellent = 4.

### Statistical analysis

IBM SPSS statistical software (version 25.0, IBM Corp) was used for statistical analyses. According to the Shapiro–Wilk test, the objective parameters and subjective scores did not exhibit the normality. Therefore, AI, noise and SNR were expressed as M [Q1, Q3], where M represents the median, and Q1 and Q3 denote the first quartile and third quartile, respectively. The differences in these parameters among various image sets were compared using the Kruskal–Wallis test with a Bonferroni *post hoc* test. The evaluation of subjective consistency between two radiologists was conducted using Cohen's kappa test, with values greater than 0.75 indicating high consistency, values ranging from 0.4 to 0.75 indicating average consistency, and values less than 0.4 indicating poor consistency. $p < 0.05$ indicates statistically significant differences.

## RESULTS

### Objective image quality evaluation

Table 1 summarized the results of noise and AI evaluations of various reconstruction algorithms and scanning scenarios (radiation doses and tube voltages). Compared to ASIR-V, the background noise of DLIR-H was reduced by 25.95% to 53.50% (paired calculation), whereas the background noise of DLIR-H MAR was reduced by 27.73% to 54.24% compared to ASIR-V MAR (paired calculation) across different dose levels and tube voltages (all $p < 0.001$). The noise of low-dose images with DLIR were similar to those of standard-dose images with ASIR-V, although there were statistically significant differences ($p < 0.05$, Table 1). Furthermore, for different tube voltages, the noise values of 70 kVp images were significantly higher than those of the other two tube voltages ($p < 0.001$), except for DLIR-H ($p = 0.87$) and DLIR-H MAR ($p = 0.735$) images at 0.5 mSv. The background noise values showed no statistically significant difference between MAR and non-MAR images across different tube voltages and radiation doses (Figs. 2A–2F). Under the same tube voltage and reconstruction algorithm for different radiation doses, the noise of the 3 mSv image group is lower than that of the 0.5 mSv group (all $p < 0.001$, File S1).

AI decreased significantly with MAR (MAR: 27.3–46.1 HU; without MAR: 60.7–115.6 HU; all $p < 0.001$) and with high tube voltages (except for 0.5 mSv ASIR-V MAR and DLIR-H MAR, all $p < 0.001$), as shown in Table 1 and Figs. 2G–2L. Compared to low-kVp images, the AI values for high-kVp images decreased except 0.5 mSv ASIR-V MAR and

**Table 1   Background noise and AI analysis across four different reconstruction algorithms (DLIR, DLIR-MAR, ASIR-V, and ASIR-V MAR), tube voltage (70 kVp/100 kVp/120 kVp) and dose level (3 mSv and 0.5 mSv) (Median [Q1,Q3]).**

| [mSv, kVp] | ASIR-V | ASIR-V MAR | DLIR-H | DLIR-H MAR | p |
|---|---|---|---|---|---|
| | | | **Noise** | | |
| [3, 70] | 32.9 [32.1, 35.1][*,+] | 35.5 [33.0, 36.8][*,+] | 17.8 [17.2, 18.9][+] | 18.9 [18.1, 20.2][*,+] | <0.001 |
| [3, 100] | 28.4 [28.1, 29.1] | 32.5 [30.4, 33.4] | 15.7 [15.2, 16.0][x] | 17.0 [16.2, 17.9] | <0.001 |
| [3, 120] | 29.9 [29.6, 31.0] | 30.5 [29.6, 32.2] | 17.2 [16.1, 17.7] | 16.6 [16.1, 17.4] | <0.001 |
| p | <0.001 | <0.001 | 0.001 | <0.001 | |
| [0.5, 70] | 67.4 [62.6, 68.9][*,+] | 63.8 [62.6, 66.0][*,+] | 37.3 [36.5, 38.3] | 38.8 [37.0, 40.2] | <0.001 |
| [0.5, 100] | 59.9 [58.0, 63.4] | 58.4 [56.8, 60.4] | 36.4 [35.1, 40.8] | 37.3 [36.3, 39.5] | <0.001 |
| [0.5, 120] | 60.9 [58.4, 63.1] | 60.6 [58.6, 62.6] | 38.2 [34.6, 40.8] | 38.2 [36.0, 40.3] | <0.001 |
| p | <0.001 | <0.001 | 0.87 | 0.735 | |
| | | | **AI** | | |
| [3, 70] | 103.5 [85.1, 219.3][*,+] | 39.4 [35.0, 48.3][*,+] | 92.4 [81.4,111.0][*,+] | 31.9 [30.2, 38.4][*,+] | <0.001 |
| [3, 100] | 115.0 [68.6, 221.5][X] | 29.1 [24.6, 35.6] | 72.6 [59.4, 94.3][X] | 27.9 [25.0, 33.2] | <0.001 |
| [3, 120] | 75.7 [58.7, 99.4] | 29.2 [25.8, 34.9] | 60.7 [50.3, 66.5] | 27.3 [23.3, 31.4] | <0.001 |
| p | 0.002 | <0.001 | <0.001 | 0.001 | |
| [0.5, 70] | 113.0 [98.8, 124.0][*,+] | 46.1 [33.5, 54.9] | 115.6 [104.3, 124.3][*,+] | 43.1 [36.6, 48.3] | <0.001 |
| [0.5, 100] | 67.8 [61.7, 77.7] | 38.3 [20.7, 50.9] | 79.6 [68.8, 83.5] | 39.7 [29.8, 46.5] | <0.001 |
| [0.5, 120] | 65.3 [56.2, 78.6] | 38.2 [4.5, 44.5] | 71.2 [66.5, 80.7] | 37.3 [28.7, 46.0] | <0.001 |
| p | <0.001 | 0.078 | <0.001 | 0.149 | |

**Notes.**

AI, artifact index; ASIR-V, 50% adaptive statistical iterative reconstruction-V; ASIR-V MAR, ASIR-V 50% with MAR; DLIR-H, deep learning image reconstruction with high strength; DLIR-H MAR, DLIR-H with MAR.

[*]Value was statistically different between 70 kVp and 120 kVp group.

[+]Value was statistically different between 70 kVp and 100 kVp group.

[X]Value was statistically different between 100 kVp and 120 kVp group.

DLIR-H MAR (*e.g.*, ASIR-V and 0.5 mSv 70/100/120 kVp: 113.0 [98.8, 124.0]/67.8 [61.7, 77.7]/65.3 [56.2, 78.6] HU), with the lowest AI value obtained using the high-kVp combined with the MAR algorithm (3 mSv 120 kVp in DLIR-H MAR: 27.3 [23.3, 31.4] HU; 0.5 mSv 120 kVp in DLIR-H MAR: 37.3 [28.7, 46.0] HU). Furthermore, there was no statistically significant difference in AI values between ASIR-V and DLIR-H, or between ASIR-V MAR and DLIR-H MAR , among four groups comparison (Kruskal–Wallis test among ASIR-V, ASIR-V MAR, DLIR-H and DLIR-H MAR) (Figs. 2G–2L). These results indicate that the AI of low-dose images with 120 kVp DLIR-H MAR was comparable to that of standard-dose images with 70 kVp ASIR-V MAR ($p = 0.143$). Under different radiation doses, the AI of images in the 3 mSv group was significantly lower than that of the 0.5 mSv group across multiple protocols (70 kVp DLIR-H, DLIR-H MAR; 100 kVp ASIR-V, DLIR-H MAR; 120 kVp ASIR-V MAR, DLIR-H, DLIR-H MAR; $p < 0.001$, File S2).

Figure 3 and Table 2 indicate that among the four reconstruction algorithms, only the 70 kVp 3 mSv protocol demonstrated statistically significant difference in SNR for artifact-impaired PNs ($p = 0.041$), whereas no significant SNR differences were observed in other comparative analyses. In the 70 kVp 3 mSv group, ASIR-V MAR showed a median SNR increase of approximately 74.5% compared to ASIR-V, while DLIR-H MAR exhibited a 137.3% higher median SNR than DLIR-H. Under the same reconstruction algorithm

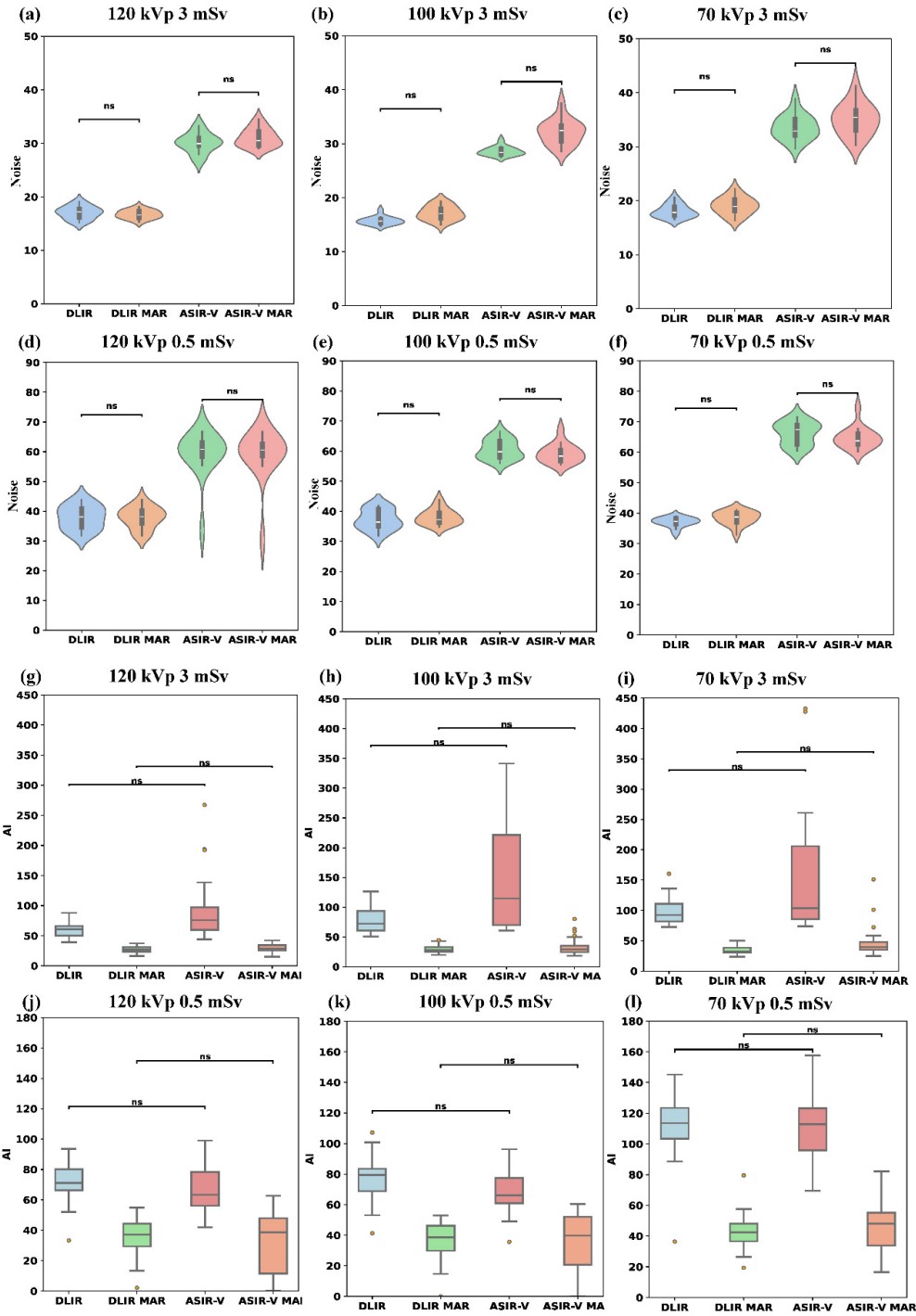

**Figure 2  Comparison of noise and AI across four reconstruction algorithms (DLIR, DLIR-MAR, ASIR-V, and ASIR-V MAR).** The labels (A–F) denote the noise levels of the four groups under the following six scanning conditions: 120 kVp 3 mSv, 100 kVp 3 mSv, 70 kVp 3 mSv, 120 kVp 0.5 mSv, 100 kVp 0.5 mSv, and 70 kVp 0.5 mSv. 

**Figure 2 (...continued)**
The labels (G–I) denote the AI levels of the four groups under the following six scanning conditions: 120 kVp 3 mSv, 100 kVp 3 mSv, 70 kVp 3 mSv, 120 kVp 0.5 mSv, 100 kVp 0.5 mSv, and 70 kVp 0.5 mSv. Ns means no statistical difference. AI, artifact index; ASIR-V, 50% adaptive statistical iterative reconstruction-V; ASIR-V MAR, ASIR-V 50% with MAR; DLIR-H, deep learning image reconstruction with high strength; DLIR-H MAR, DLIR-H with MAR.

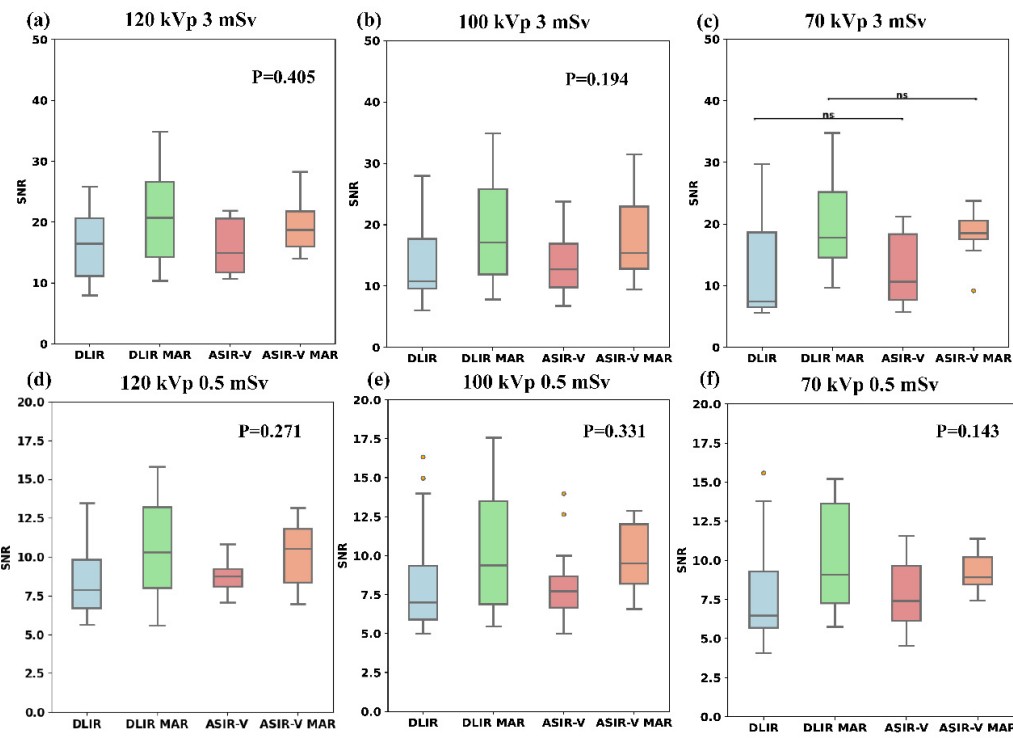

**Figure 3** **Boxplots of SNR across four reconstruction algorithms.** Boxplots of SNR across four reconstruction algorithms (DLIR, DLIR-MAR, ASIR-V, and ASIR-V MAR). The labels (A–F) denote the SNR levels of the four groups under the following six scanning conditions: 120 kVp 3 mSv, 100 kVp 3 mSv, 70 kVp 3 mSv, 120 kVp 0.5 mSv, 100 kVp 0.5 mSv, and 70 kVp 0.5 mSv. ASIR-V, 50% adaptive statistical iterative reconstruction-V; ASIR-V MAR, ASIR-V 50% with MAR; DLIR-H, deep learning image reconstruction with high strength; DLIR-H MAR, DLIR-H with MAR.

and radiation dose, there was no statistical difference in the SNR of artifact- impaired PNs among three tube voltages (*p*-values refer to Table 2). Furthermore, except for the 70 kVp ASIR-V and 70 kVp DLIR-H groups, the SNR for artifact-impaired PNs in the 3 mSv groups was significantly higher than that of the 0.5 mSv groups across all other scan protocols (with varying voltages and reconstruction algorithms), demonstrating statistical significance ($p < 0.05$).

Regarding the NPS, the trends of the NPS area under various kVp levels, radiation doses, and reconstruction algorithms were similar to those of background noise. Higher kVp, higher doses, or DLIR were more likely associated with the lower NPS areas. Regarding noise texture, the differences in $f_{peak}/f_{avg}$ values between MAR and non-MAR images were

**Table 2** SNR of artifact impaired non-solid PNs across four different reconstruction algorithms (DLIR, DLIR-MAR, ASIR-V, and ASIR-V MAR), tube voltage (70 kVp/100 kVp/120 kVp) and dose level (3 mSv and 0.5 mSv) (Median [Q1,Q3]).

| [mSv, kVp] | ASIR-V | ASIR-V MAR | DLIR-H | DLIR-H MAR | p |
|---|---|---|---|---|---|
| | | | SNR | | |
| [3, 70] | 10.6 [7.7, 18.4] | 18.5 [17.5, 20.5] | 7.5 [6.5, 18.6] | 17.8 [14.5, 25.2] | 0.041 |
| [3, 100] | 12.7 [9.8, 16.9] | 15.4 [12.8, 23.0] | 10.8 [9.6, 17.7] | 17.1 [11.9, 25.8] | 0.194 |
| [3, 120] | 14.9 [11.7, 20.6] | 18.7 [16.0, 21.8] | 16.5 [11.1, 20.7] | 20.7 [14.2, 26.6] | 0.405 |
| p | 0.190 | 0.641 | 0.199 | 0.879 | |
| [0.5, 70] | 7.4 [6.1, 9.6] | 8.9 [8.5, 10.2] | 6.47 [5.69, 9.30] | 9.1 [7.3, 13.6] | 0.143 |
| [0.5, 100] | 7.7 [6.7, 8.7] | 9.5 [8.2, 12.0] | 7.0 [5.9, 9.4] | 9.4 [6.9, 13.5] | 0.331 |
| [0.5, 120] | 8.8 [8.1, 9.21] | 10.5 [8.4, 11.8] | 7.9 [6.7, 9.8] | 10.3 [8.0, 13.2] | 0.271 |
| p | 0.462 | 0.574 | 0.543 | 0.911 | |

**Notes.**

SNR, signal-to-noise ratio; ASIR-V, 50% adaptive statistical iterative reconstruction-V; ASIR-V MAR, ASIR-V 50% with MAR; DLIR-H, deep learning image reconstruction with high strength; DLIR-H MAR, DLIR-H with MAR.

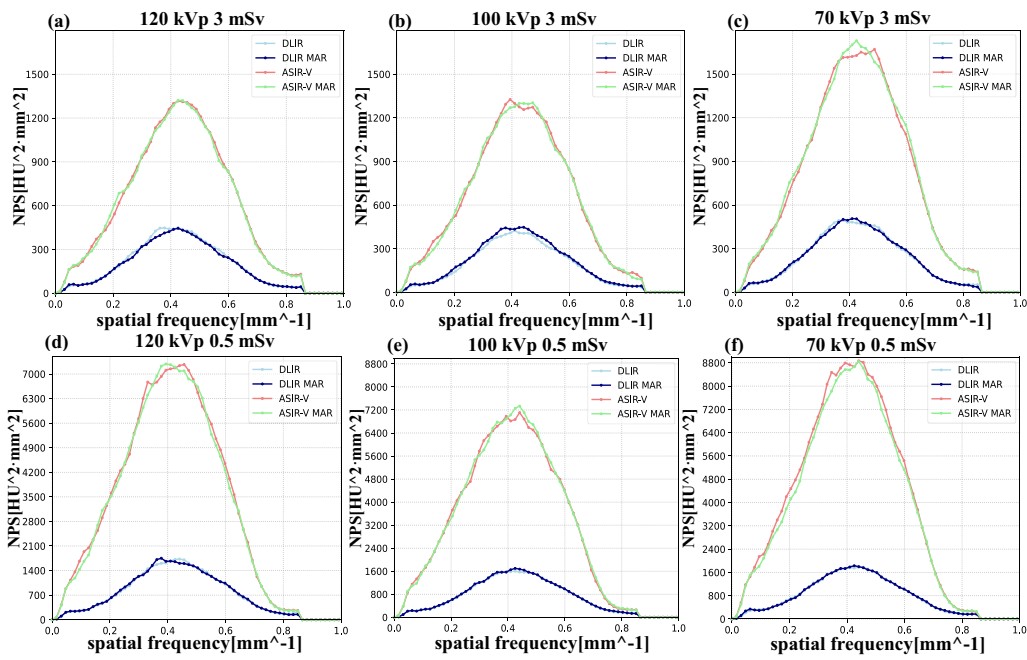

**Figure 4** NPS curves across four reconstruction algorithms (DLIR, DLIR-MAR, ASIR-V, and ASIR-V MAR). The labels (A–F) denote the NPS curves of the four groups under the following six scanning conditions: 120 kVp 3 mSv, 100 kVp 3 mSv, 70 kVp 3 mSv, 120 kVp 0.5 mSv, 100 kVp 0.5 mSv, and 70 kVp 0.5 mSv. obtained at various scanning scenarios. ASIR-V, 50% adaptive statistical iterative reconstruction-V; ASIR-V MAR, ASIR-V 50% with MAR; DLIR-H, deep learning image reconstruction with high strength; DLIR-H MAR, DLIR-H with MAR.

insignificant, and both were closer to the reference values (3 mSv, FBP). Similarly, the influences of different voltages and radiation doses on $f_{peak}/f_{avg}$ were also negligible in our study (Table 3 and Fig. 4).

**Table 3  NPS analysis across four different reconstruction algorithms (DLIR, DLIR-MAR, ASIR-V, and ASIR-V MAR), tube voltage (70 kVp/100 kVp/120 kVp) and dose level (3 mSv and 0.5 mSv).**

| [mSv, kVp] | ASIR-V | ASIR-V MAR | DLIR-H | DLIR-H MAR | Ref (3 mSv, FBP) |
|---|---|---|---|---|---|
| NPS area (HU$^2$ mm) | | | | | |
| [3, 70] | 41.7 | 41.6 | 22.0 | 22.1 | 77.7 |
| [3, 100] | 35.9 | 36.8 | 20.0 | 20.3 | 66.2 |
| [3, 120] | 36.4 | 36.6 | 20.4 | 20.3 | 67.1 |
| [0.5, 70] | 93.8 | 93.4 | 41.3 | 41.3 | 77.7 |
| [0.5, 100] | 85.1 | 84.4 | 39.7 | 39.7 | 66.2 |
| [0.5, 120] | 84.8 | 85.0 | 40.4 | 40.2 | 67.1 |
| $f_{peak}$ (mm$^{-1}$) | | | | | |
| [3, 70] | 0.44 | 0.42 | 0.39 | 0.40 | 0.45 |
| [3, 100] | 0.42 | 0.45 | 0.40 | 0.41 | 0.45 |
| [3, 120] | 0.45 | 0.43 | 0.40 | 0.42 | 0.43 |
| [0.5, 70] | 0.42 | 0.43 | 0.43 | 0.42 | 0.45 |
| [0.5, 100] | 0.42 | 0.42 | 0.42 | 0.42 | 0.45 |
| [0.5, 120] | 0.43 | 0.41 | 0.43 | 0.41 | 0.43 |
| $f_{avg}$ (mm$^{-1}$) | | | | | |
| [3, 70] | 0.43 | 0.42 | 0.43 | 0.42 | 0.43 |
| [3, 100] | 0.43 | 0.43 | 0.43 | 0.43 | 0.44 |
| [3, 120] | 0.43 | 0.43 | 0.43 | 0.43 | 0.44 |
| [0.5, 70] | 0.41 | 0.41 | 0.43 | 0.43 | 0.43 |
| [0.5, 100] | 0.41 | 0.41 | 0.43 | 0.43 | 0.44 |
| [0.5, 120] | 0.41 | 0.41 | 0.43 | 0.43 | 0.44 |

**Notes.**

NPS, noise power spectrum; ASIR-V, 50% adaptive statistical iterative reconstruction-V; ASIR-V MAR, ASIR-V 50% with MAR; DLIR-H, deep learning image reconstruction with high strength; DLIR-H MAR, DLIR-H with MAR; $f_{peak}$, the peak spatial frequency of NPS; $f_{avg}$, the average spatial frequency of NPS.

## Qualitative analysis

The interobserver agreements were significant concerning the extent of metal artifacts, the confidence in diagnosing artifact-impaired PNs, and the overall image quality ($\kappa = 0.79$, 0.82, 0.78 for 3 mSv, all $p < 0.001$; 0.74, 0.74, 0.73 for 0.5 mSv, all $p < 0.001$). The median [Q1, Q3] of the extent of metal artifacts assessments for 0.5 mSv ASIR-V, ASIR-V MAR, DLIR-H, and DLIR-H MAR were as follows: 2 [2, 2], 3 [3, 3], 2 [2, 3], and 3 [3, 3] ($p < 0.001$), and for 3 mSv were as follows: 2 [2, 3], 4 [4, 4], 2 [2, 3], and 4 [4, 4] ($p < 0.001$). The influence of artifacts in images with MAR was significantly lower than those in images without MAR, aligning with the findings from the objective evaluation. The median [Q1, Q3] of the overall image quality assessments for 0.5 mSv ASIR-V, ASIR-V MAR, DLIR-H, and DLIR-H MAR were as follows: 2 [1, 3], 3 [3, 3], 2 [2, 3], and 3 [3, 3] ($p < 0.001$), while for 3 mSv, the values were 2 [1, 3], 4 [3, 4], 2 [1.75, 3], and 4 [4, 4] ($p < 0.001$). These results indicate that DLIR-H MAR/ASIR-V MAR exhibited superior image quality compared to DLIR-H/ASIR-V ($p < 0.001$). In terms of diagnostic confidence for PNs, there were statistically significant differences in the subjective scores among different algorithms,

**Table 4** Subjective score across four different reconstruction algorithms (DLIR, DLIR-MAR, ASIR-V, and ASIR-V MAR), tube voltage (70 kVp/100 kVp/120 kVp) and dose level (3 mSv and 0.5 mSv).

| [mSv, kVp] | ASIR-V | ASIR-V MAR | DLIR-H | DLIR-H MAR |
|---|---|---|---|---|
| | **The extent of metal artifact** | | | |
| [3, 70] | 2 | 3.67 | 2 | 4 |
| [3, 100] | 2 | 4 | 2 | 4 |
| [3, 120] | 2.83 | 4 | 3.17 | 4 |
| [0.5, 70] | 1.83 | 3 | 2 | 3 |
| [0.5, 100] | 2 | 3 | 2 | 3 |
| [0.5, 120] | 2.5 | 3 | 3 | 3.5 |
| | **The confidence of PNs diagnosis** | | | |
| [3,70] | 2 | 3 | 2 | 4 |
| [3, 100] | 2.33 | 4 | 3 | 4 |
| [3, 120] | 3 | 4 | 3 | 4.17 |
| [0.5, 70] | 2 | 3 | 2 | 3 |
| [0.5, 100] | 2 | 3 | 2 | 3 |
| [0.5, 120] | 2.5 | 3 | 3 | 3 |
| | **The overall image quality** | | | |
| [3, 70] | 1 | 3 | 1.33 | 4 |
| [3, 100] | 2 | 4 | 2 | 4 |
| [3, 120] | 3 | 4 | 3 | 4 |
| [0.5, 70] | 1 | 3 | 2 | 3 |
| [0.5, 100] | 2 | 3 | 2 | 3 |
| [0.5, 120] | 3 | 3 | 3 | 3.5 |

the median [Q1, Q3] for 0.5 mSv ASIR-V, ASIR-V MAR, DLIR-H, and DLIR-H MAR were as follows: 1 [1, 2], 3 [2, 3], 2 [1, 2], and 4 [2.75, 4] ($p < 0.001$), while for 3 mSv, the values were 2 [1.75, 3], 4 [3.5, 4], 3 [3, 3], and 4 [4, 4] ($p < 0.001$), indicating that metal artifacts in the images significantly affected the diagnostic confidence for PNs lesions under the influence of artifacts (Table 4).

## DISCUSSION

In this study, we assessed the performance of the combinations of MAR and DLIR algorithms under various scanning scenarios on artifact reduction and image quality improvement. Both objective and subjective analyses showed that the MAR algorithm combined with DLIR-H at 120 kVp could significantly reduce metal artifacts and improve image quality while preserving image texture in artifact-free regions.

As demonstrated in Fig. 2 and Table 1, noise levels were predominantly influenced by the utilization of the DLIR algorithm and radiation dose, whereas MAR algorithms, designed for artifact suppression, demonstrated negligible impact on noise characteristics. Regarding the DLIR algorithm, previous literature has demonstrated its potential to improve image quality (*Li et al., 2024*), enhance diagnostic confidence (*Zhu et al., 2024*), and reduce radiation dose (*Yang et al., 2021*; *Jiang et al., 2022a*), particularly in the detection of lung nodules. *Jiang et al. (2022a)* demonstrated the feasibility of using DLIR for lung

nodule screening with chest X-ray doses, showing that images at 0.07/0.14 mSv yielded comparable results in lung nodule detection, SNR, and malignant features to those of 3 mSv enhanced images. *Zhao et al. (2023)* presented superior accuracy and repeatability in the detection of pulmonary lesions and nodules based on DLIR (*D'Hondt et al., 2024*). In line with these studies (*Szczykutowicz et al., 2021*), our research also showed that DLIR maintained lower noise levels than ASIR-V, regardless of tube voltages or radiation dose levels (all $p < 0.001$). The noise of low-dose images with DLIR-H were comparable to those of standard-dose images with ASIR-V, although there were statistically significant differences ($p < 0.05$, Table 1). For different tube voltages, under identical reconstruction algorithms and radiation dose conditions, the 70 kVp protocol exhibited significantly higher noise compared to 100 kVp/120 kVp settings in all groups ($p < 0.001$) except the 0.5 mSv DLIR-H (0.87) and DLIR-H MAR (0.735). This observation may be explained by the characteristics of metal-implanted scans: higher kVp settings improve photon penetration efficiency, allowing the detector to capture more effective signals, thereby reducing noise in reconstructed images.

For the AI results, previous studies have demonstrated that the MAR algorithm alone can reduce metal artifacts to improve the delineation accuracy in dental implants (*Fukugawa et al., 2022*) or diagnostic confidence in knee implant (*Zhang et al., 2020*). In line with these papers, our study demonstrated that AI values in MAR images were significantly lower than those in images without MAR across various tube voltages, radiation dose levels and reconstruction algorithms (all $p < 0.001$). Furthermore, the Bonferroni *post hoc* tests of AI between DLIR-H and ASIR-V groups (or DLIR-H MAR VS ASIR-V MAR) were not statistically significant among the four-groups comparison: ASIR-V, ASIR-V MAR, DLIR-H and DLIR-H MAR. *Kovacs et al. (2018)* reported similar results. However, when only compared the AI between ASIR-V and DLIR-H images (or DLIR-H MAR VS ASIR-V MAR) using Mann–Whitney test across various scenarios, DLIR-H reconstructed images exhibited lower AI than those of ASIR-V, and differences were statistically significant at 3 mSv without MAR algorithm across various tube voltages. The $p$-values were 0.001, 0.003, and 0.036 for 120, 100, and 70 kVp, respectively. The lack of statistical differences between the ASIR-V and DLIR-H subgroups (Bonferroni *post hoc* test of Kruskal–Wallis test) in the multi-group comparison (Kruskal–Wallis test) may result from the pronounced differences between images with and without MAR, which obscured the AI differences between DLIR-H and ASIR-V. Regarding the tube voltage in metal artifact reduction, some studies have demonstrated that high keV of spectral CT combined with the MAR algorithm effectively reduces artifacts and improves diagnostic confidence (*Chae et al., 2020*), while other studies have indicated that moderate keV (70–80 keV) with the MAR algorithm has the best trade-off between vascular clarity and artifact levels (*Zhao et al., 2023*). In our study, the 70 kVp protocol produced significantly higher AI values compared to other tube voltages ($p < 0.001$) in all groups except the 0.5 mSv ASIR-V MAR ($p = 0.078$) and DLIR-H MAR ($p = 0.149$). The lack of statistically significant differences in AI values between different tube voltages within these two low-dose groups may be attributed to the combined effects of increased noise from low radiation doses and the effective artifact reduction by MAR algorithms, which collectively diminished the inherent advantages of

higher kVp in metal artifact reduction. Furthermore, the AI of 120 kVp images with DLIR-H MAR at 0.5 mSv was comparable to this of 70 kVp images with ASIR-V MAR at 3 mSv ($p = 0.143$). Thus, DLIR-H combined with high kVp and MAR algorithm allows for the possibility of low-dose scanning in the context of metal implants. Qualitative evaluations of the extent of metal artifact by observers demonstrated significant concordance with quantitative metrics.

For most of the SNR values for artifact-impaired PNs, there were no statistical differences among the four groups of reconstruction algorithms (except for 3 mSv, 70 kVp). This may be because the degree of artifact impact on the four nodules varied, resulting in a wide range of SNR values under the same algorithm. Therefore, even though the differences in medians were substantial, there was no statistical significance. However, considering the statistical results of the 70 kVp 3 mSv group and the median SNR values of the other groups, the SNR values in the MAR algorithm groups are higher than that in the no-MAR algorithm groups (ASIR-V *vs.* ASIR-V MAR or DLIR-H *vs.* DLIR-H MAR). The increase in SNR is primarily attributed to two factors: an increase in CT values or a decrease in SD values. The reduction in SD values is mainly due to the decrease in artifacts and noise. In non-enhanced images, the advantages of high contrast and CT values in low keV images are diminished due to the absence of iodine contrast agents. Thus, decreased SD is the primary contributor to the increased SNR. Results showed that the noise value of MAR images is comparable to non-MAR images; however, the AI value decreases significantly in MAR images, leading to a lower SD. Thus, due to the metal artifact reduction, images with MAR had the higher SNR compared to images without MAR.

Furthermore, to the best of our knowledge, our study is the first to introduce the NPS for evaluating whether the MAR algorithm alters image texture in artifact-free regions. The results indicated that image texture was not affected by the choice of tube voltage, dose levels and reconstruction algorithm with or without MAR.

This study has several limitations: first, this study was performed on a phantom, which inherently lacks the anatomical complexity and did not account for the clinical diversity of patients. Future clinical studies with large sample sizes are warranted to investigate the clinical efficacy of combining MAR with DLIR in enhancing diagnostic accuracy and artifact suppression capability in CT images with metal implants. Second, to assess the ability of metal artifact suppression, three tube voltages were selected, but 140 kVp was not among them. Third, the metrics related to PNs, particularly volume, were not assessed in this study, because volume measurements by the AI software were inaccurate due to interference from metal artifacts and cotton. Finally, the exclusive investigation of GE's proprietary reconstruction algorithms (DLIR and ASIR-V) inherently limits the generalizability of our conclusions to other vendor platforms.

## CONCLUSION

In conclusion, the combination of the MAR algorithm with DLIR-H demonstrated significant noise and AI reductions, SNR improvements, while preserving the image texture of artifact-free regions. Interestingly, the low-dose images reconstructed by DLIR-H MAR

at 120 kVp had comparable noise and AI compared to standard-dose images by ASIR-V MAR at 70 kVp.

## ACKNOWLEDGEMENTS

We appreciate the CT scanning technology support provided by GE Corporation.

### Funding

The authors received no funding for this work.

### Competing Interests

Mengya Guo is currently an employee of GE HealthCare, the manufacturer of the CT system used in this study. GE HealthCare had no input regarding the study data or analysis. The other authors have no conflicts of interest to declare.

### Author Contributions

- Huachun Zou performed the experiments, analyzed the data, prepared figures and/or tables, authored or reviewed drafts of the article, and approved the final draft.
- Zonghuo Wang performed the experiments, prepared figures and/or tables, and approved the final draft.
- Mengya Guo performed the experiments, analyzed the data, prepared figures and/or tables, and approved the final draft.
- Kun Peng performed the experiments, prepared figures and/or tables, and approved the final draft.
- Jian Zhou performed the experiments, prepared figures and/or tables, and approved the final draft.
- Lili Zhou conceived and designed the experiments, analyzed the data, authored or reviewed drafts of the article, and approved the final draft.
- Bing Fan conceived and designed the experiments, analyzed the data, authored or reviewed drafts of the article, and approved the final draft.

### Data Availability

The raw data is available in the Supplemental Files.

### Supplemental Information

Supplemental information for this article can be found online at http://dx.doi.org/10.7717/peerj.19516#supplemental-information.

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
