# Peer review of "Metal artifact reduction combined with deep learning image reconstruction algorithm for CT image quality optimization: a phantom study"

_PeerJ, doi:10.7717/peerj.19516_

## Round 0.1 · original submission · Major Revisions

Address all the comments in a revision and appropriate rebuttal

Reviewer 1 ·

Basic reporting

The manuscript is well-structured and presents promising results, but some areas need improvement. The language could be clearer, especially for an international audience, and the introduction should better highlight the study's novelty. Figures and tables require more detailed captions, and the discussion should explicitly link results to hypotheses. Additionally, the study's limitation of using a phantom should be acknowledged, with suggestions for future clinical research. These improvements will enhance the manuscript's clarity and impact.

Experimental design

The study is relevant and well-conducted, but the introduction should more clearly highlight the knowledge gap and novelty of combining MAR with DLIR. The methods section needs more detail to ensure reproducibility, particularly regarding reconstruction parameters and ROI selection. Additionally, future studies should include larger sample sizes for more robust statistical analysis. These improvements will enhance the manuscript's clarity and impact.

Validity of the findings

The findings are supported by quantitative and qualitative analyses, enhancing their validity. However, the use of a phantom limits generalizability to clinical settings. The authors should acknowledge this limitation and suggest future research with human subjects. Additionally, some statistical analyses, such as SNR comparisons, were not performed due to small sample size, which may affect the robustness of the results. Future studies should include larger sample sizes for more comprehensive statistical analysis.

Additional comments

1. Figure 1 is labeled with (a) to (e), but the specific content of each subfigure is not clearly explained in the figure caption or the main text.
2. The manuscript utilizes reconstruction algorithms such as ASIR-V, ASIR-V MAR, DLIR-H, and DLIR-H MAR, but does not provide a brief introduction to these methods. Readers may not fully understand the specific principles and differences between these algorithms, which could affect their comprehension of the research methodology. Please include a brief explanation of the basic principles and distinctions of ASIR-V, ASIR-V MAR, DLIR-H, and DLIR-H MAR in the Methods section.
3. The study repeated scans only 3 times per condition, resulting in a small sample size that may limit the power of statistical analysis and the reliability of the results. It is recommended to increase the number of scans (e.g., 5-10 times) to enhance the statistical power and to explicitly address the limitations of the small sample size in the discussion. Although the current study provides valuable results through non-parametric tests, increasing the sample size and improving statistical methods would further strengthen the scientific rigor and reliability of the research.

Reviewer 2 ·

Basic reporting

This paper evaluates the efficacy of the metal artifact reduction (MAR) algorithm in eliminating metal artifacts and its influence on image quality under different scanning parameters. Specifically, a phantom with a pacemaker was scanned at different doses and voltages. The reconstruction results of adaptive statistical iterative reconstruction-V (ASIR-V), ASIR-V with MAR, high-strength deep learning image reconstruction (DLIR-H), and DLIR-H with MAR were compared. Finally, the paper concluded that the MAR algorithm combined with DLIR-H can reduce metal artifacts and enhance the overall image quality, particularly at high kVp tube voltages. The detailed comments are provided as follows:

Experimental design

The paper lacks innovation, and its conclusions are evident. The conclusion that the MAR algorithm can reduce metal artifacts has been widely verified in the literature of various MAR algorithms. Additionally, the conclusion that image quality improves with high kVp tube voltages, which is due to the weaker beam hardening effect at high kVp, has been verified in other studies.
 The experiment involved scanning only a single phantom with a pacemaker, resulting in a lack of diverse metal materials. It was difficult to obtain reliable conclusions due to the limited sample size. To obtain more robust results, it is recommended to use a variety of phantoms and scan them in the presence of different metal objects.

Validity of the findings

The paper compares only one iterative algorithm and one deep learning algorithm. Given that different manufacturers and devices employ varying algorithms, the conclusions appear to be applicable solely to the CT equipment used in the experiment, precluding the derivation of general conclusions.
 Although the Lungman chest phantom contains numerous anatomical structures, significant differences remain between it and real patients. Consequently, relying solely on this phantom study is insufficient to achieve the author's goal of identifying the optimal protocol for clinical application. The impact of different scanning protocols on image quality should be further evaluated with clinical research.
 Please provide detailed descriptions or literature sources for the algorithms utilized in the article, including the DLIR-H, ASIR-V, and MAR algorithms.
 In the experiment, the pacemaker was positioned on the surface of the phantom for scanning. Given the flexible installation of the pacemaker, it is advisable to scan the phantom both with and without metal separately. This approach allows for the accurate measurement of metal artifacts by comparing the differences between the two scans.
 The experiment involved repeating the scan three times (line 111). If there are any differences between each scan, please provide a detailed description. If there are no differences, please explain the specific purpose of repeating the scans, as there is no obvious difference in the images obtained from each scan.
 The title is too long, as "Metal artifact reduction" and "reduce metal artifacts" convey similar meanings. It is essential to optimize the title for brevity.
 Figs. 2 and 4 are blurry, please replace them with clearer images.

Reviewer 3 ·

Basic reporting

Paper entitled “ Metal artifact reduction combined with deep learning image reconstruction algorithm to reduce metal artifacts and improve image quality-A phantom study(#109684)”


General overview: This paper evaluates the impact of combining the MAR algorithm with different reconstruction methods, including ASIR and DLIR, across various tube potentials and dose levels, using both quantitative and qualitative analyses. Notably, this study not only assesses artifact extent and noise levels in artifact-affected regions but also examines image texture and quality through NPS comparisons in artifact-free areas and comprehensive quality assessments.

General comments:
Authors should replace 'AI' with 'MAI' or another suitable term to avoid confusion, as 'AI' is commonly associated with artificial intelligence. Additionally, the results lack a thorough comparison between ASIR-V MAR and DLIR-H MAR in terms of artifact index and noise.In the introduction section, please provide a general overview of metal artifacts in CT imaging and the techniques used for metal artifact reduction (MAR).Throughout the entire manuscript, ensure that the significance level and p-value are reported wherever statistical significance is mentioned. This will provide readers with clear, transparent information on the statistical analysis and strengthen the credibility of the findings. The discussion section should follow the same order as the results are presented. For example, if noise results are discussed first in the results section, they should be addressed first in the discussion section to maintain a clear and logical flow. There are several areas of ambiguity throughout the manuscript that need to be addressed, and I have pointed these out specifically in my comments

Experimental design

Abstract:
1. Please ensure that the abstract adheres to the journal's formatting guidelines. I recommend structuring it using the following headings: Background, Methods, Results, and Discussion.
2. Purpose: Line 21: Please specify the type of MAR used in this study to ensure clarity and reproducibility.
3. Results: Please include the results of the NPS analysis in the abstract to ensure a comprehensive summary
4. In lines 35-40, please report the statistical significance when presenting the results to enhance clarity and scientific rigor.

Introduction:

1. The introduction currently begins with a description of CIED, resulting in an abrupt start. To create a smoother transition, in the first paragraph, I recommend first providing a brief overview of what metal artifact is in CT imaging, followed by their impact on images containing CIED, briefly explain current MAR strategies. This will help establish context before discussing CIED. Additionally, I suggest citing the following paper to give readers the opportunity to explore Metal Artifact Reduction (MAR) techniques in detail:
Gjesteby L, De Man B, Jin Y, Paganetti H, Verburg J, Giantsoudi D, Wang G. Metal artifact reduction in CT: Where are we after four decades? IEEE Access. 2016 Sep 13;4:5826-49.

2. Line 53: Since MAR+VMI represents an acquisition-based improvement to MAR, it is best to differentiate it from other MAR algorithms. I suggest making this distinction clearer by revising the text accordingly. Please replace “the artifact-reduction effects of the metal artifact reduction (MAR) algorithm, particularly when combined with high keV virtual monoenergetic images (VMI) derived from spectral CT” with “The artifact-reduction effect of various metal artifact reduction (MAR) algorithms (e.g., projection completion MAR, iterative MAR), particularly when combined with high-keV virtual monoenergetic images (VMI) derived from spectral CT as an acquisition-based improvement to MAR..”

3. These are some highly cited papers on MAW+VMI, I suggest citing those:
Khodarahmi I, Haroun RR, Lee M, Fung GSK, Fuld MK, Schon LC, et al. Metal artifact reduction computed tomography of arthroplasty implants: effects of combined modeled iterative reconstruction and dual-energy virtual monoenergetic extrapolation at higher photon energies. Invest Radiol. 2018;53:728–735. doi: 10.1097/RLI.0000000000000497.
Bongers MN, Schabel C, Thomas C, Raupach R, Notohamiprodjo M, Nikolaou K, Bamberg F. Comparison and combination of dual-energy-and iterative-based metal artefact reduction on hip prosthesis and dental implants. PLoS One. 2015 Nov 24;10(11):e0143584.
Laukamp KR, Zopfs D, Lennartz S, Pennig L, Maintz D, Borggrefe J, et al. Metal artifacts in patients with large dental implants and bridges: combination of metal artifact reduction algorithms and virtual monoenergetic images provides an approach to handle even strongest artifacts. Eur Radiol. 2019
Long Z, DeLone DR, Kotsenas AL, Lehman VT, Nagelschneider AA, Michalak GJ, Fletcher JG, McCollough CH, Yu L. Clinical assessment of metal artifact reduction methods in dual-energy CT examinations of instrumented spines. American Journal of Roentgenology. 2019 Feb;212(2):395-401.)

4. Line 72 needs citations.
5. Lines 75 and 77 need citations
6. Line 83: Please specify the type of MAR algorithm used in the study, such as whether it is an iterative MAR or a projection completion MAR, to ensure clarity and reproducibility.
7. Line 83: I suggest stating "pacemakers (a specific type of CIED)" to clarify the relevance of CIED information in the introduction section.
8. Line 84: I understand that "tube voltage" and "tube potential" are often used interchangeably in published papers. However, for greater precision, please use "tube potential" throughout the manuscript, as voltage is a unit rather than a quantity.

Methods and materials:

1. Line 142: Please specify that M[Q1, Q3] refers to the mean (M) along with the first quartile (Q1) and third quartile (Q3),
2. Statistical analysis: Please clarify why statistical analysis could not be performed on the SNR. Since the measurements were taken from the same phantom, wouldn’t the number of repeats and other conditions for SNR measurements be consistent with those for AI? This would suggest that statistical analysis should be feasible for both.
3. Please report size of ROI.
Results:
1. According to the journal guidelines, all statistical parameters should be reported, including the corresponding test statistic, degrees of freedom, the exact p-value (rather than general statements like "p<0.05"), and effect sizes. Please ensure that this is consistently applied throughout the manuscript.
2. Line 152: Since Table 1 summarizes the results of noise and artifact index, it would be incorrect to say, "Table 1 summarized the results of all the objective evaluations." Please revise this statement to accurately reflect the contents of Table 1. For example, you could say, "Table 1 summarizes the results of noise and artifact index evaluations."
3. Lines 154: It appears there is a discrepancy between the reported percentage reduction in background noise for DLIR-H compared to ASIR-V and the values presented in Table 1. Based on my calculations, the noise reduction for DLIR-H compared to ASIR-V falls between 43% and 76%, rather than the 36.13%-57.69% range stated in the manuscript.
(my calculations: For the highest range of noise change: the lowest noise of DLIR-H across different reconstructions and techniques (15.7) minus the highest noise of ASIR-V across different reconstructions and techniques (67.4), divided by the highest noise of ASIR-V (67.4), gives a noise reduction of 76% (i.e., (15.7 - 67.4) / 67.4 = 0.76 = 76%).
• For the lowest range of noise change: the highest noise of DLIR-H across different reconstructions and techniques (38.2) minus the highest noise of ASIR-V (67.4), divided by the highest noise of ASIR-V (67.4), gives a noise reduction of 43% (i.e., (38.2 - 67.4) / 67.4 = 0.43 = 43%). Even when comparing the noise of each DLIR and ASIR at the same technique, the lowest and highest noise reductions I calculated would still range from 37% to 63%.If there has been a misunderstanding in the interpretation or calculation of the data, please clarify the results in the manuscript to ensure they accurately reflect the findings. This may involve revising the percentage reductions or explaining how the values in Table 1 were derived, especially if the methodology for calculating noise changes differs from your approach.


4. Line 156: It appears that the p-value you referenced is not reported in Table 1. In Table 1, you compare different reconstruction methods within the same technique (mSv, kV), and also compare different techniques while using the same reconstruction methods. However, there is no comparison across different reconstruction methods and different techniques. Please consider adding those.

5. Lines 157-158: Please avoid generalizing the results. Based on Table 1, DLIR-H and DLIR-MAR appear to be exceptions, so it's important to present their results clearly. I suggest revising lines 157-158 as follows: “Furthermore, the noise values of 70 kVp images were significantly higher than those of the other two tube voltages, except for DLIR-H and DLIR-MAR images at 0.5 mSv. Additionally, the noise value at 3 mSv and 70 kV was significantly different from the 100 kV value only in DLIR-H images.”
6. Lines 159-160: all statistical results should be fully reported
7. Please also comment on the differences in noise values between DLIR-H MAR and ASIR-V MAR across the different protocols.
8. Line 161: Please avoid using qualitative statements such as "pronounced improvement" in the results section. Instead, use precise quantitative language. For example, you could revise the sentence as follows: "In the MAR algorithms, AI decreased significantly across different reconstruction methods, tube potentials, and radiation doses (MAR: 27.3-46.1 HU; without MAR: 60.7-115.6 HU; all p < 0.001)."
9. Line 163: Please cite table 1 and fig2 g-l
10. Line 163-165: “Compared to low-kVp images, the AI values for high-kVp images were decreased (e.g., ASIR-V and 0.5 mSv 70/100/120 kVp: 113.0 [98.8, 124.0]/67.8 [61.7, 77.7]/65.3 [56.2, 78.6]).” , Please avoid generalizing the results, as not all AI values at low kVs significantly increased compared to other kVs. Instead, specify the exceptions clearly.
11. Line 164: decreased instead of “were decreased”.
12. Line 168: Please report the p-value for the significance of the difference between ASIR-V and DLIR. Additionally, what about the difference between DLIR-MAR and ASIR-MAR? Is it significant? Please ensure that all relevant p-values are included for clarity and accuracy.
13. Line 169: Avoid using expressions like "interestingly." Instead, present the information in a more neutral, objective manner. For example, you could rephrase sentences to focus on the findings directly, such as:"These results indicate that..." or "It was observed that..."

14. Line 172: Data is given for 3mSv, 100 kV; please replace 100 with 120 kV.
15. Lines 172-174, Is the increase in SNR for MAR-related images statistically significant when compared to non-MAR images? Please check the data and include the corresponding p-value to confirm whether the difference is significant.
16. Lines 174-176: IIt is important to always state if the changes were statistically significant. Please include the significance for the increase in SNR with an increase in tube potential for Non-MAR images, specifying the p-value and whether the change is statistically significant.
17. Line 177: Please move any interpretation of the results to the discussion section. The statement, "Thus, the MAR algorithm served as the primary factor for artifact mitigation compared to kVp," seems to be a conclusion, which would be better suited for the discussion section. Additionally, for further context and support, you may want to refer to the following paper with similar findings: Reynoso-Mejia, Carlos A., Jonathan Troville, Martin G. Wagner, Bernice Hoppel, Fred T. Lee Jr, and Timothy P. Szczykutowicz. "Needle artifact reduction during interventional CT procedures using a silver filter." BMC Biomedical Engineering 6, no. 1 (2024): 2.
18. Line 183: Please state that reference values is 3mSv, Filtered back projection.
19. Line 186: Avoid using expression words such as “substantial” in results.
20. Lines 187-188, Please ensure that the p-value, along with other statistical information such as the test statistic, degrees of freedom, and effect size, is reported in the table for clarity and completeness. This will help readers better understand the statistical significance of your findings.
21. Please clarify if these values (mean[Q1, Q3]) refer to each dose level of reconstruction. It would be helpful to rephrase as: "The mean[Q1, Q3] for each dose level and reconstruction method are as follows..."This will provide clear context and ensure that the data is easily interpreted.
22. Line 192: Throughout the entire manuscript, please ensure that you report the significance level and p-value wherever you mention statistical significance. This will provide readers with clear, transparent information on the statistical analysis and strengthen the credibility of your findings.
23. Do the same thing as comment 21 for 194-195.
24. Lines 197-199 should be moved to discussion
25. Give results for diagnostic confidence in PNS

Discussion:
1. Line 224: I propose adding the following to explain why the SD of MAR is lower:
" Thus decreased SD is the primary contributor to the increased SNR. Results showed that the noise value of MAR images is comparable to non-MAR images; however, the AI value decreases significantly in MAR images, leading to a lower SD. Thus,..."This will provide a clearer rationale for the lower SD in MAR images.
2. Lines 236-239: Albeit, the reduction in noise for DLIR vs ASiR is already well understood, you still add to this field by adding mar. Please see and cite the following paper for comaprison for the characterization of DLIR and ASIR: Szczykutowicz, Timothy P., Brian Nett, Lusik Cherkezyan, Myron Pozniak, Jie Tang, Meghan G. Lubner, and Jiang Hsieh. "Protocol optimization considerations for implementing deep learning CT reconstruction." American Journal of Roentgenology 216, no. 6 (2021): 1668-1677.

3. Line 237-239, If the results are statistically different, we cannot describe them as comparable. Instead, we should clearly state the observed statistical difference.
4. Lines 241-242; Please clarify what you mean by "among four group comparison." It would be helpful to specify which groups are being compared to ensure the context is clear for the reader.
5. Lines 243-247: Results should not be introduced for the first time in the discussion section. Moreover, the result here contradicts what was presented in the results section. The results indicated that there was no statistically significant difference between the AI values of DLIR and ASIR images. However, in this section, it is stated that the difference was significant.
6. Lines 247-249, “The lack of statistical differences between the ASIR-V and DLIR subgroups in the multi-group comparison may result from the pronounced differences between images with and without MAR, which obscured the AI differences between DLIR and ASIR-V.” "What do you mean by 'subgroups' and 'multi-groups'? This paragraph is unclear in its current form."
7. Lines 249-251:This sentence is a repetition of the result and should be omitted.
8. "Although several studies have evaluated the effect of different MAR techniques on NPS and noise texture, some of which have focused on artifact regions, to the best of our knowledge, our study is the first to assess NPS in artifact-free regions ". Pleas cite this as one of those doing NPS and feel free to add any additional papers evaluating NPS that you find relevant.)

Validity of the findings

Tables:
1. Tables should be self-explanatory. Please specify 'Mean[Q1,Q3]' in the table legend. The headings 'Noise' and 'AI' should be placed in a more appropriate location; currently, they are under 'mSv' and 'kV,' which is confusing. It would be clearer to position them directly below the 'Reconstruction Algorithms.' Additionally, please specify in the table legend what each p-value corresponds to."
2. For table 2, state in the legend that this is SNR of artifact impaired non-solid PNS. Please move heading in the right columns.
3. Table 3: Move the headings to the appropriate location. Please report the p-values and update the table legend to specify NPS, fpeak, and fave analysis. Ensure that all p-values and statistical analyses are reported.

4. Table 4: Move headings to the right location. Report p value and statistical analysis.

Figures:
1. Fig. 1 Please specify in the table legend that the red arrows indicate PNs.
2. Fig. 2 Please state the p-value over each ns in the figure. In figures a-f, specify the significance of the difference between each DLIR vs ASIR and DLIR-MAR vs ASIR-MAR. In figures g-l, state the significance of the difference between each MAR and non-MAR image.
3. Figure 3: Please specify what each p-value corresponds to in the figure and state this clearly in the legend. The legend should fully explain the contents of the figure, ensuring that all comparisons and statistical details are clearly outlined
4. Figure 4: Please use the recommended resolution for PeerJ, as the current image resolution is too low, making it difficult to distinguish between different colors and notes. Additionally, consider using a color-blind tool to select more accessible color choices

Additional comments

I would like to acknowledge the author's hard work, and the effort put into gathering and presenting the data. However, after a thorough review, I believe that the manuscript requires some major revisions before it can be considered for publication in this journal. I have provided detailed feedback and suggestions in order to help improve the manuscript's quality and clarity.

---

## Round 0.2 · accepted · Accept

The manuscript has been considerably improved following a thorough revision in accordance with the reviewers' comments and now meets the journal's standards for publication.

Reviewer 1 ·

Basic reporting

The author has adequately responded to the majority of the issues raised in my initial feedback.

Experimental design

None.

Validity of the findings

None.

Additional comments

None.

Reviewer 3 ·

Basic reporting

The authors responded to and edited the comments I suggested.

Experimental design

The authors responded to and edited the comments I suggested.

Validity of the findings

The authors responded to and edited the comments I suggested.